# CryoET reveals actin filaments within platelet microtubules

Chisato Tsuji [1], Marston Bradshaw[2], Megan F. Allen[2], Molly L. Jackson [2], Judith Mantell[1], Ufuk Borucu[3], Alastair W. Poole[2], Paul Verkade [1], Ingeborg Hers[2] ✉, Danielle M. Paul[2] ✉ & Mark P. Dodding [1] ✉

Crosstalk between the actin and microtubule cytoskeletons is important for many cellular processes. Recent studies have shown that microtubules and F-actin can assemble to form a composite structure where F-actin occupies the microtubule lumen. Whether these cytoskeletal hybrids exist in physiological settings and how they are formed is unclear. Here, we show that the short-crossover Class I actin filament previously identified inside microtubules in human HAP1 cells is cofilin-bound F-actin. Lumenal F-actin can be reconstituted in vitro, but cofilin is not essential. Moreover, actin filaments with both cofilin-bound and canonical morphologies reside within human platelet microtubules under physiological conditions. We propose that stress placed upon the microtubule network during motor-driven microtubule looping and sliding may facilitate the incorporation of actin into microtubules.

Crosstalk between the actin and microtubule cytoskeletons is important for many cellular processes, including cell division, migration and intracellular transport[1,2]. This is thought to be mediated by proteins that directly connect or signal between the two dynamic networks to coordinate their activities. However, recent studies have provided evidence for an additional mechanism for crosstalk, by showing that microtubules and F-actin can assemble to form a composite structure where F-actin occupies the lumen of the cylindrical tubulin polymer[3,4].

Lumenal actin filaments were identified in human HAP1 cells treated with a small molecule that targets kinesin-1 to induce the formation of thin, membrane-bound, microtubule-based projections that are highly accessible to cryo-electron tomography (cryoET)[4]. In this system, microtubules form a dynamic bundle and lumenal actin filaments are highly abundant, with morphologies (named Class I and Class II) that are distinct from the canonical cytoplasmic/muscle form; principally, the cross-over spacing between the two 'long-pitch' strands of the actin double helix is short (at around 27 nm), compared to the canonical 35–37 nm[5–7]. This suggested that actin-binding proteins (ABPs) which modify the twist of the filament may be present[8,9].

CryoET analysis of *Drosophila* S2 cells revealed similar filaments in microtubule-based projections in cells treated with the actin-targeting drug cytochalasin D, whose abundance was enhanced by additional treatment of cells with thapsigargin[3]. Sub-tomogram averaging and knockdown studies demonstrated that these filaments are composed of cofilin-bound F-actin (cofilactin); cofilin is known to change the twist of F-actin[9,10].

Together, these studies show that F-actin can occupy the microtubule lumen in chemically-induced microtubule-based projections emerging from human and insect cells in culture. However, how these structures are formed is unknown. Moreover, the impact of these small-molecule manipulations and the fact that these studies were performed on cultured cells, leaves the physiological occurrence of lumenal actin unclear.

Here, we examine the composition of lumenal F-actin in human HAP1 cells and show that the Class I filament is composed of cofilin-bound F-actin. To explore the requirements for the formation of these structures, we utilise an in vitro reconstitution system which demonstrates that, although cofilin-bound F-actin can be incorporated into microtubules, cofilin is not essential. A dynamic and bundled

[1]School of Biochemistry, Faculty of Health and Life Sciences, Biomedical Sciences Building, University Walk, University of Bristol, BS8 1TD Bristol, UK. [2]School of Physiology, Pharmacology and Neuroscience, Faculty of Health and Life Sciences, Biomedical Sciences Building, University Walk, University of Bristol, BS8 1TD Bristol, UK. [3]GW4 Facility for High-Resolution Electron Cryo-Microscopy, University of Bristol, Bristol, UK. ✉e-mail: i.hers@bristol.ac.uk; danielle.paul@bristol.ac.uk; mark.dodding@bristol.ac.uk

microtubule cytoskeleton is present in native human platelets[11] and Focussed-Ion-Beam (FIB) milling and cryoET reveals that actin filaments with both cofilin-bound and canonical morphologies are found within platelet microtubules, providing an unequivocal identification of lumenal F-actin in a physiological setting that is not modified by small-molecule treatment.

## Results

### Characterisation of lumenal F-actin in HAP1 cell protrusions

The cofilin-bound F-actin recently identified within the microtubule lumen in *Drosophila* S2 cells is morphologically similar to the Class I filament observed in human HAP1 cells[3,4] There are two shared features of particular note. Firstly, the short spacing of the 'crossovers' of the actin double helix that can be readily identified and directly measured in high-quality tomograms and are apparent as layer lines in their power spectra[4]; and secondly, the smooth appearance of the filament that is distinct from the classic 'beads on a string' appearance of canonical F-actin[12]. To determine whether human Class I filaments are cofilin-bound F-actin, we treated HAP1 cells with the kinesin-1 targeting small-molecule 'kinesore' to induce projections using our established protocol and imaged those projections with cryoET using a 300 kV Titan Krios microscope (Fig. 1a–c)[4,13]. In this new dataset, we observed lumenal actin filaments corresponding to the Class I and Class II forms (Fig. 1b and d, top left)[4]. Class II filaments showed an F-actin-like power spectrum with short crossover spacing augmented with a prominent meridional layer line indicating a deviation from the typical F-actin helical morphology (Supplementary Fig. 1a). We also observed rarer examples of a morphologically distinct filament that we named Class III that did not display an F-actin-like power spectrum (Fig. 1d, top right, Supplementary Fig. 1b). Nonetheless, images showing transitions between filament classes through breaks or less well-defined intermediates suggest that each of these filaments may be actin structures, perhaps undergoing disassembly or assembly, or with different binding partners (Fig. 1d, middle and bottom). This opens the intriguing possibility that lumenal actin is dynamic.

Of 65 lumenal filaments in this new dataset (Fig. 1e), 40 displayed the Class I morphology, with a crossover spacing (measured directly from layer lines) of 27.66 nm (+/− 0.23 s.d.) that contrasted with the long cross-over spacing of a canonical actin filament (Fig. 1f, g). We performed helical reconstructions to generate 3D maps on the five highest-quality images, and one representative example is shown (Fig. 1h). Docking of a cryo-electron microscopy (cryoEM) structure of canonical F-actin (light blue) (PDB: 8D17)[14] into the Class I helical reconstruction model was insufficient to explain the density. In contrast, when a cryoEM cofilin-actin structure (dark blue and magenta) (PDB: 3J0S)[15] was docked in, it fitted our model well (cross-correlation score 0.96) (Fig. 1h). Similarly, our model fitted well with density of cofilin-actin filaments from the lumen of microtubules in S2 cells (Supplementary Fig. 1c)[3]. Together, these data strongly support the proposition that the Class I filament in HAP1 cells is cofilin-bound F-actin. Whilst the identity of the cofilin family member is unclear, prior transcriptomic analysis suggests that Cofilin 1 (CFN1) is the main form expressed in this cell type[16].

### In vitro reconstitution of lumenal F-actin and cofilin-bound F-actin

To test this proposition further and to determine whether cofilin is essential for the formation of lumenal actin, we used an in vitro reconstitution system, taking advantage of the 'TicTac' buffer that allows dynamic assembly of both polymers[17] (Fig. 2). When purified actin and tubulin were polymerised together, both F-actin and microtubules could be observed using cryoET (Fig. 2a). In samples where cofilin (CFN1) was present, the short crossovers and distinct morphology of the actin filaments was readily apparent (pink arrowheads)[9]. This suggested that these characteristics are a good proxy for the presence of high levels of bound cofilin in cryoET images of F-actin in situ (Fig. 2b) and provide further support for our findings in HAP1 cells. Occasionally, F-actin and cofilin-bound filaments were found to run alongside microtubules, but most formed a loose mesh that was not microtubule-associated. In both conditions, we were able to observe lumenal filaments in a minority of the microtubules (2% for F-actin alone and 5% for cofilin-actin) (Fig. 2c, d, Supplementary Fig. 2). Although this difference was not statistically significant ($p = 0.1$), it is consistent with the greater abundance of cofilin-actin observed in situ, and may point to a role for cofilin in the formation and/or stabilisation of these structures. The appearance of in vitro lumenal cofilin-actin filaments was indistinguishable from Class I filaments observed in situ in HAP1 cells. We also noted examples of actin/cofilin-actin filaments apparently associated with breaks in the microtubule lattice or emerging from microtubule ends (Fig. 2e, f). Thus, both F-actin and cofilin-bound F-actin can be incorporated into the microtubule lumen. Although cofilin-bound F-actin is the predominant form in HAP1 and S2 cells (pointing to a functional or regulatory role for cofilin), addition of cofilin is not essential for F-actin incorporation in this simple in vitro system.

### Identification of lumenal F-actin in human platelets

We next sought to establish whether lumenal actin filaments exist in native or primary cells under physiological conditions without small-molecule treatments. We considered that one factor in common between the cell-based systems where microtubule lumenal F-actin has been unambiguously observed is that microtubules are most likely under significant mechanical stress, as the formation of extended microtubule-based projections is driven by microtubule motors[4,18]. Indeed, projections in HAP1 cells form through the extrusion of tight microtubule loops that push against the plasma membrane[4]. CryoET analysis of the tips of these loops shows evidence of extreme microtubule curvature, breakage and depolymerisation and the presence of all three filament classes (both in and outside of the microtubule lumen) (Supplementary Fig. 3; Supplementary Movie 1). Ostensibly similar motor-driven microtubule looping is important for extrusion in platelet biogenesis from mega-karyocytes[19] resulting in a dynamic circumferential bundle of microtubules within platelets themselves. This characteristic bundle, called the marginal band, mechanically maintains the platelet discoid shape during their resting state and rapidly remodels under platelet activation[11,20–22].

Because of the relative thickness of platelets (compared to small-molecule-induced projections), we turned to FIB-milling. Human platelets were put on EM grids, plunge frozen, thinned by FIB-milling and analysed using cryoET (Fig. 3a). We observed bundled microtubule structures which are consistent with the marginal band[23–25] (Fig. 3b; Supplementary Movie 2), as well as more isolated microtubules (Fig. 3c, d). Within the microtubules, actin filaments were identified in 6 out of 28 tomograms with visible sections of microtubule lumen from 2 independent platelet preparations. Consistent with previous observations in HAP1 and S2 cells, the majority had the distinctive Class I cofilin-bound F-actin morphology (Fig. 3b, c insets; Fig. 3e, and Supplementary Fig. 4a). An example of a Class I filament in the platelets was used to perform helical reconstruction, and when a cryoEM cofilin-actin structure was docked in, it fit the model well (correlation score 0.93) (Fig. 3f). We also observed less frequent examples of long-crossover filaments more closely resembling the canonical cofilin-free form, consistent with our in vitro reconstitution assays (Fig. 3d inset).

Platelet microtubules have been reported to contain 13-15 protofilaments[24,26]. To ask whether the presence of a lumenal filament is associated with a difference in protofilament number we performed rotational averaging of microtubules from our platelet tomographic reconstructions[12]. In both cases ($n = 10$), most microtubules (70–80%) were composed of 13 protofilaments, although we also observed 14 protofilament microtubules. Thus in platelets, and

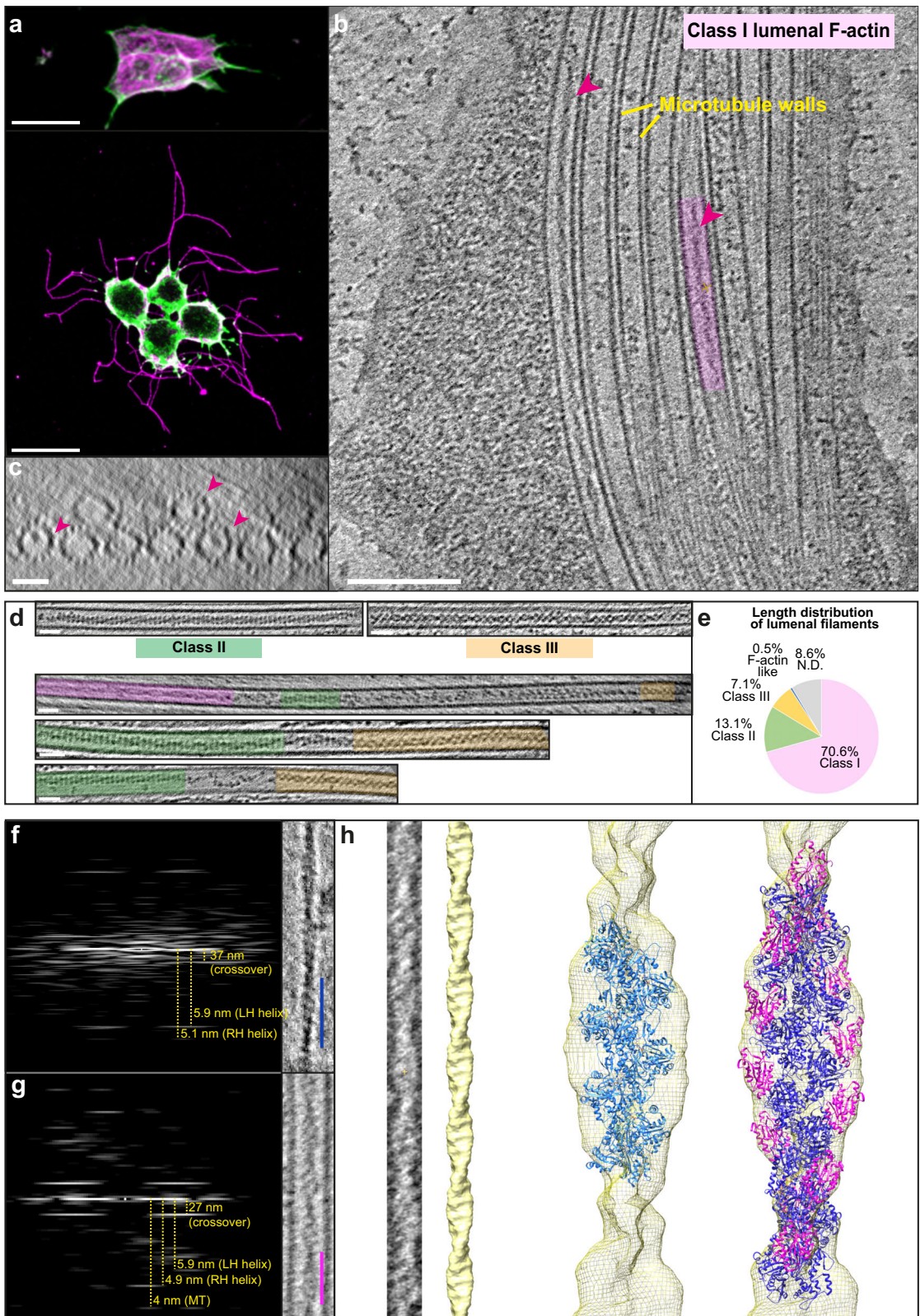

consistent with Santos et al.[3], presence of lumenal filaments does not appear to correlate with protofilament number (Supplementary Fig. 4b, c, d).

We did not observe clear examples of Class II and Class III filaments in platelets or in vitro. This may reflect the relatively high abundance of lumenal filaments in the HAP1 cell system or a missing component in the in vitro reconstitution system. In any case, the presence of these structures in HAP1 cell projections highlights the importance of considering intermediates on an assembly/disassembly pathway. This may be particularly important when examining morphologically diverse lumenal material in other cell types that has filamentous character but unknown composition[27]. Structural studies of Class II and Class III filaments are, therefore, a priority.

**Fig. 1 | Morphology and composition of lumenal actin filaments.**
**a** Representative fluorescence microscopy images (*n* > 3) showing control (top) and kinesore-treated (bottom)) HAP1 cells stained using antibodies targeting beta-tubulin (magenta) and actin (green). Scale bar = 20 µm. **b, c** Tomogram slices (in longitudinal (**b**) (Scale bar = 100 nm) and transverse orientations (**c**) (Scale bar = 25 nm)) showing microtubules within a projection from a kinesore-treated HAP1 cell. Class I lumenal filaments are highlighted with magenta arrows and one example is boxed in magenta. Images here and below are representative of >3 independent freezing and imaging sessions. **d** Examples of Class II and Class III filaments (top) and transitions between filament morphologies (middle and bottom). Class I is shaded magenta, Class II is green and Class III is orange (Scale bar = 25 nm). **e** Distribution of the length of filaments found in the microtubule lumen by their class. The total filament length was 22.7 µm which is 27.4% of the total microtubule length (82.7 µm) across 144 microtubules in 26 tomograms. **f** Layer lines from an in vitro example (blue scale bar = 37 nm) of a canonical F-actin labelled with real space distances annotated. This is representative of >10 filaments. **g** Layer lines of an example class I filament (from HAP1 cells) (magenta scale bar = 27 nm) labelled with real space distances, representative of >10 filaments **h** An example of a segment of class I filament that has been straightened, inverted, and projected in z, which was averaged to produce a helical reconstruction map (EMD-50845). This is representative of 5 similar reconstructions. When structures of actin (PDB: 8D17, light blue) was docked in, it was insufficient to fill the map, whereas cofilin-actin (PDB: 3JOS, actin in dark blue, cofilin in magenta) is in good agreement with the model.

## Discussion

In summary, we have characterised the form and composition of microtubule lumenal F-actin in situ, reconstituted cofilin-free and cofilin-bound forms in vitro, and demonstrated that lumenal F-actin is present in ex vivo primary human cells without chemical modification. Going forward, it will be useful to explore the effect of mutations in cytoskeletal genes that cause hereditary thrombocytopenia, including in the cofilin pathway, for effects on lumenal F-actin[28–30]. More broadly, the identification of lumenal F-actin in this physiological setting should lead to a sustained effort to understand the function and regulation of lumenal actin in platelets and determine its presence in other cell types. It will be important to consider factors that may control its incorporation or function, such as microtubule posttranslational modifications. Our observations suggest that these efforts should focus on settings where microtubules are under structural and mechanical stress.

There are now many high-resolution imaging studies examining microtubules in their native environment that have not observed lumenal filaments but that have observed extensive particulate and globular lumenal material, for example, in neurons[31,32] and malaria parasites[33,34], as well as others reviewed in ref. 27. However, filamentous lumenal structures have recently been observed in fibroblasts grown in cell-derived matrices[35]. It will be important to determine what is F- or G-actin in these diverse cell types, what other proteins are present in the lumen, and if and how they interact with lumenal actin. Our findings suggest that a full mechanistic understanding will require the development of new models of single filament actin dynamics in confined environments and new experimental systems to explore the impact of lumenal F-actin on microtubule properties.

## Methods

### HAP1 cell culture and fluorescence imaging

HAP1 cells were grown in Iscove-modified Dulbecco Media with 10% FBS and penicillin/streptomycin at 37 °C in a 5% CO2 incubator. For fluorescence imaging, cells were methanol fixed and stained using TUB 2.1 monoclonal antibody (1:1000, T4026, Sigma-Aldrich) detected with anti-mouse secondary antibody directly conjugated to Alexa Fluor 568 (1:500, A11004, Thermo Fisher Scientific), and stained for β-actin using 13E5 rabbit monoclonal antibody directly conjugated to Alexa Fluor 488 (1:500, 8844, Cell Signalling Technology)[4].

### HAP1 cell cryoET

HAP1 cells were cultured as above. For kinesore treatment, Quantifoil R1.2/1.3 gold 300 mesh grids (Agar Scientific) were coated with 1 mg/ml of fibronectin in a 6-well plate overnight at 37 °C. Cells were then plated at $0.5 \times 10^5$ cells per well and incubated for 2 days at 37 °C. Cells were washed with Ringer's buffer (pH 6.8) and treated with kinesore (Cambridge Corporation) at 0.2% concentration in DMSO (0.2% DMSO for control), in Ringer's buffer (pH 6.8) for 1 hour, in a non-CO2 incubator. Grids were lifted out of the media and 10 nm gold fiducial markers (Sigma-Aldrich) were applied on the grids.

Samples were blotted and plunge-frozen in liquid ethane using a Leica EM GP plunge freezer. Cryo-EM grids were clipped and screened on Tecnai20 LaB6 TEM (FEI) at 200 kV with a Gatan 626 cryo-transfer holder and sent to Diamond Light Source Electron Bio-Imaging Centre (eBIC) for tomography data collection at 64,000x magnification on a 300 kV Titan Krios microscope with Falcon4 detector and a 5 eV slit (Thermo Fisher). Tomographic series were acquired using a dose symmetric scheme, with increments of 3 degrees. The images shown in Supplementary Fig. 3 of a projection tip were acquired as part of the dataset reported in ref. 4 and processed as described there.

### Tomographic reconstruction of HAP1 projections

The IMOD packages newstack and alignframes were used to order and motion correct the tilt series as part of a custom Bash script (Dr Mathew McLaren, University of Exeter, available at https://github.com/mathewmclaren/cryoem-stuff), and reconstructed into a tomogram using weighted back projection with 5 Simultaneous Iterative Reconstruction Technique like filter on Etomo (IMOD). Reconstructions were performed using IMOD and its Etomo interface[36]. Where tomograms were used for helical reconstruction, CTF correction was performed using the Etomo 3D CTF option. The tomogram presented in Supplementary Fig. 3 was acquired as part of the dataset reported in ref. 4 and processed as described there.

### Layer line analysis and helical reconstruction

Fourier transforms of 2D projection of extracted filament volumes were performed and layer line positions were measured using Fiji (ImageJ). Filaments of interest were extracted from 3dmod to ImageJ to straighten the filaments and invert densities. Realspace helical reconstruction was performed using IMAGIC on individual filaments with a length of 512 pixels[37], with dimensions (axial rise 27.5 and subunit rotation 162°) calculated from the layer line analysis. Cofilin-actin (PDB - 3JOS) and actin (PDB - 8D17) were docked into helical reconstruction models in UCSF Chimera. A correlation score was obtained by fitting a 20 Å map generated from cofilin-actin (PDB - 3JOS) and fitting into the reconstructed model.

### In vitro reconstitution and cryoET

Tubulin (Cytoskeleton Inc: HTS03-A) and actin (Cytoskeleton Inc: APHL99) were co-polymerised at 4 mg/ml and 0.4 mg/ml respectively in a modified TicTac buffer (10 mM HEPES, 80 mM PIPES (pH 6.8), 50 mM KCl, 5 mM $MgCl_2$, 1 mM EGTA, 1 mM GTP, 2.7 mM ATP, 1 mM DTT)[17] at 37 °C for 1 hour. Cofilin (Cytoskeleton Inc: CF01-A) was added to the mixture before polymerisation at 0.2 mg/ml. 5 µl of the polymerised mixture was pipetted onto Quantifoil R1.2/1.3 copper 300 mesh grids (Agar Scientific) and plunge frozen in liquid ethane using a Leica EM GP plunge freezer. CryoEM grids were clipped and imaged on Talos Arctica (FEI) at 63,000x and a dose symmetric scheme with increments of 3 degrees from −60 to 60 degrees and reconstructed as above.

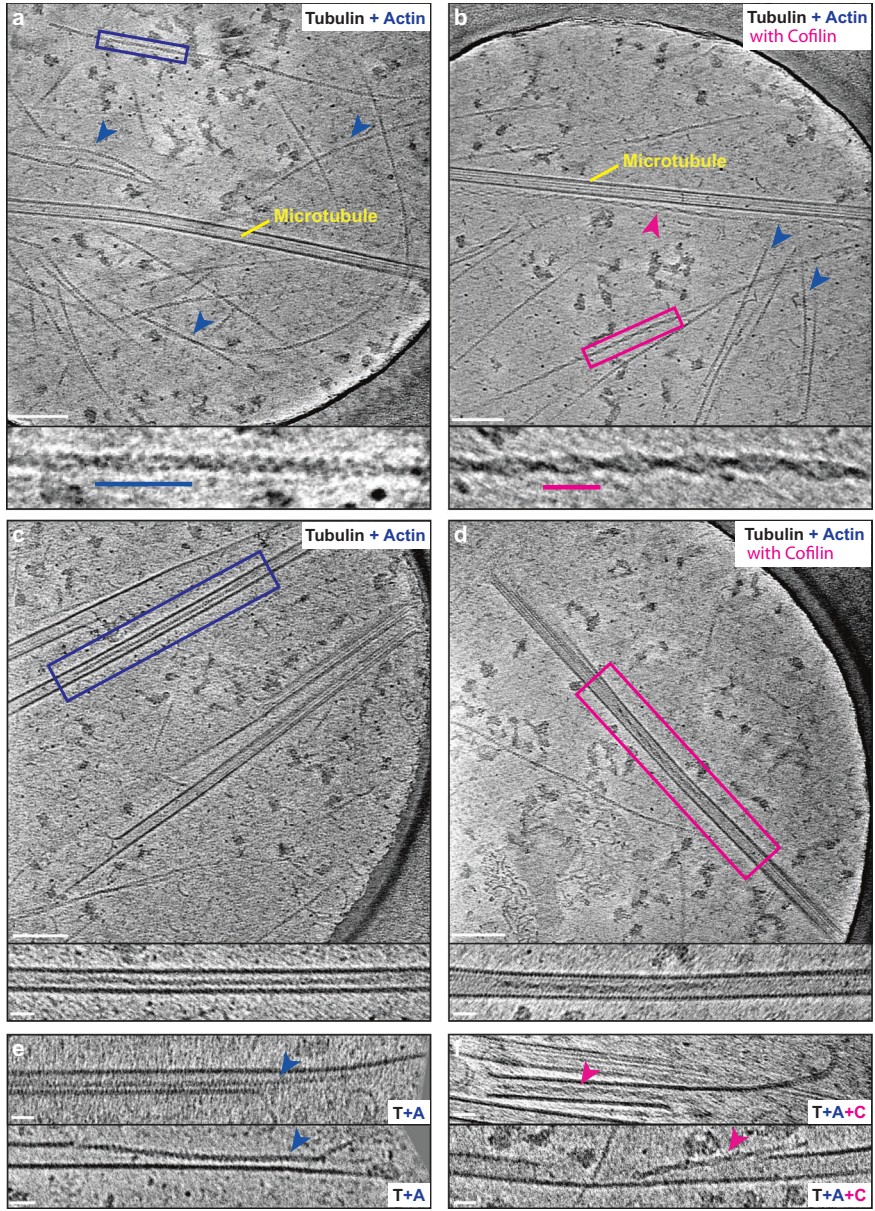

**Fig. 2 | In vitro reconstitution of lumenal F-actin and cofilin-bound F-actin.**
**a** Representative slice from a tomogram showing organisation of microtubules and actin filaments that have been polymerised together (Scale bar = 100 nm). Expanded box highlights the canonical 'beads-on-a-string' F-actin morphology and long crossover spacing (blue line is 37 nm). **b** Representative slice from a tomogram showing the typical organisation of microtubules and actin filaments that have been polymerised together in the presence of cofilin (Scale bar = 100 nm). Expanded box highlights the distinctive smooth appearance of the cofilin-bound actin filaments and short-crossover spacing (magenta line is 27 nm). **c** Representative slice from a tomogram showing a canonical actin filament within a microtubule, boxed and expanded below. Scale bar on main panel is 100 nm, and 25 nm on the zoom panel. **d** Representative slice from a tomogram showing a cofilin-bound actin filament within a microtubule, boxed and expanded in below. Scale bar on main panel is 100 nm, and 25 nm on the zoom panel. **e** Examples of actin filaments at microtubule ends or breaks in the lattice (Scale bar = 25 nm) (**f**) Examples of cofilin-bound actin-filaments at microtubule ends or breaks in the lattice (Scale bar = 25 nm). Examples shown in this figure are representative of 3 independent polymerisation and freezing sessions.

## Platelet isolation, FIB-milling, and cryoET

Approval for the platelet work in this study was granted to IH by South Central−Hampshire A Local Research Ethics Committee (NHS-REC reference 20/SC/0222), in accordance with the Declaration of Helsinki. All donors provided informed consent and signed a consent form. Platelets were isolated according to established protocols[38], as follows. Blood from healthy drug-free volunteers was drawn into 3.2% (w/v) trisodium citrate. Blood was centrifuged at $180 \times g$ 17 min at room temperature and platelet-rich plasma (PRP) was collected and supplemented with acidified Acid Citrate Dextrose (ACD) 1/7 (v/v) and apyrase (0.02 U/ml). Platelets were subsequently pelleted ($520 \times g$/10 min) and washed with CGS (13 mM trisodium citrate, 30 mM glucose, 120 mM sodium chloride) supplemented with 0.02 U/ml apyrase, before resuspension in modified HEPES-Tyrodes buffer (145 mM NaCl, 1 mM $MgCl_2$, 3 mM KCl, 10 mM HEPES pH 7.3) supplemented with 5.5 mM D-glucose and 0.02 U/ml apyrase, at a concentration of $4 \times 10^8$ platelets/ml. Quantifoil R1.2/1.3 gold 300 mesh grids (Agar Scientific) were coated with Collagen Related Peptide at 50 μg/ml overnight at room temperature and blotted off or left uncoated. We did not observe a difference between uncoated and coated samples, and so data presented are pooled sets. 5 μl of isolated platelets were pipetted onto the grids and left for 5 minutes, blotted,

and plunge frozen in liquid ethane using a Leica EM GP plunge freezer. CryoEM grids were clipped and screened on a Talos Arctica (FEI).

Focused Ion Beam milling (FIB-milling) was performed at eBIC on an Aquilos cryoFIB/SEM (Thermo Fisher) to produce lamellae at 12-

degree tilts. The samples were then imaged using a Titan Krios with a Falcon 4i detector at 300 keV at 53,000x with a dose symmetric scheme from −45 degrees to 69 degrees. The tomograms were reconstructed on the eBIC processing pipeline using AreTomo[39].

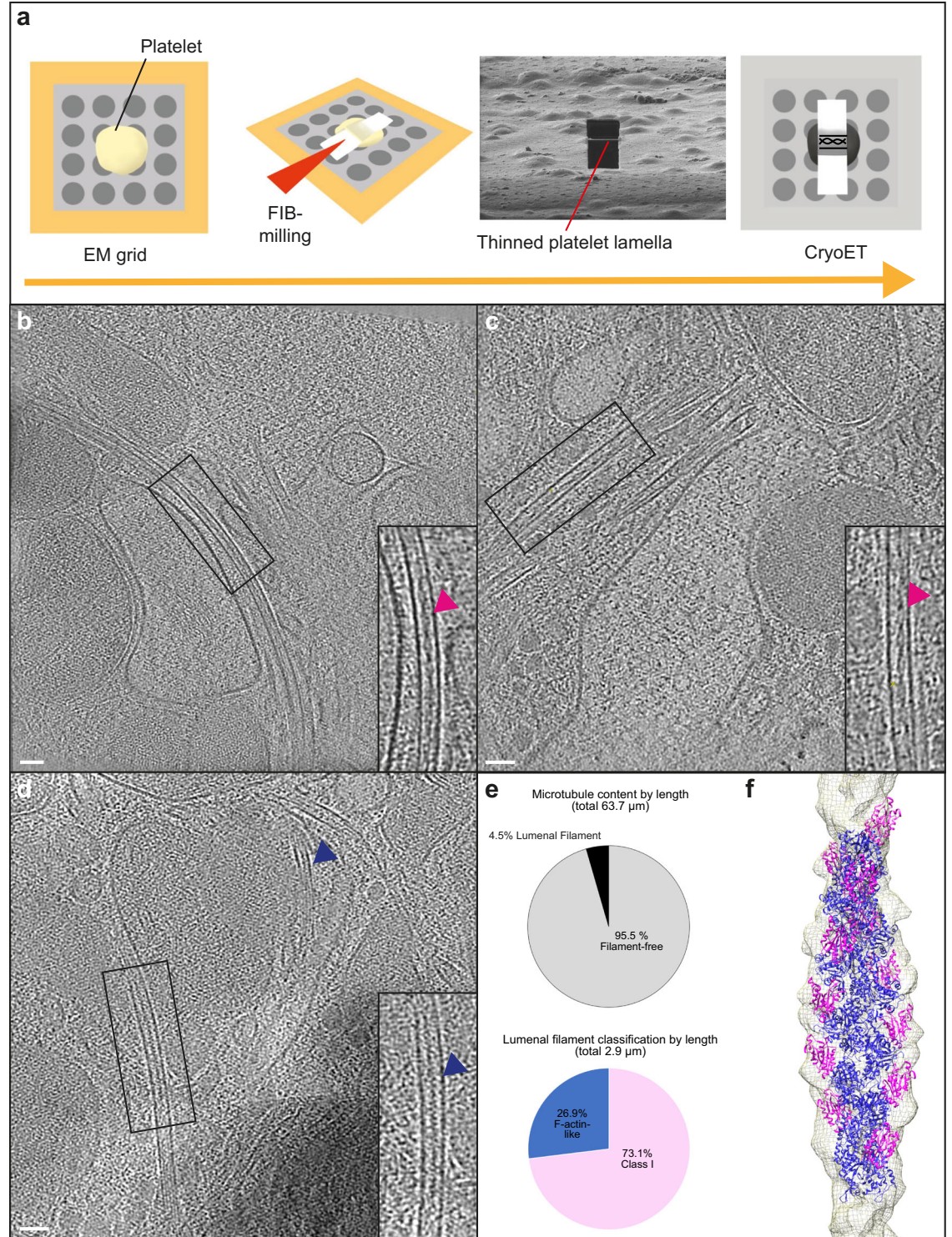

**Fig. 3 | F-actin resides within the lumen of human platelet microtubules.**
**a** Schematic showing platelet FIB-milling and cryoET workflow. **b**–**d** Representative slices from tomograms showing cofilin-bound actin filament within platelet microtubules (magenta arrows) and an actin filament with a canonical morphology (blue arrow)(Scale bars in all panels are 50 nm). **b** Is a 11-microtubule bundle. Two microtubules contain filaments with the Class I morphology, seven appear empty or contain globular densities. Contents of the remaining microtubules were ambiguous. A video showing a Z-series through (**b**) is provided in Supplementary Movie 2. Examples shown in this figure are representative of two independent platelet preparations, freezing, and FIB-milling/cryoET sessions. **e** Quantification of total microtubule length occupied by lumenal filaments in 163 microtubules from 28 tomograms, corresponding to 23 lamellae across 2 datasets. **f** A helical reconstruction map (EMD-50814) produced from a platelet tomogram with cofilin-actin docked into the model (PDB: 3JOS, actin in dark blue, cofilin in magenta).

## Protofilament number for platelet microtubules

To determine the protofilament number for platelet microtubules, we performed rotational averaging on straight segments of microtubules, using C13, 14 or 15 symmetry (IMAGIC)[12,37]. The cross-sectional views, as well as the 3D model viewed on UCSF Chimera, were used to identify the correct number of protofilaments for the 10 microtubules containing filaments, as well as 10 microtubules without filaments in the same tomograms.

## Reporting summary

Further information on research design is available in the Nature Portfolio Reporting Summary linked to this article.

## Data availability

Helical reconstruction maps are deposited in the Electron Microscopy Data Bank with accession code EMD-50845 for the HAP1 helical reconstruction and EMD-50814 for the platelet helical reconstruction. Source data are provided in this paper.

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

## Acknowledgements

C.T. is supported by the Wellcome Trust 4-year PhD Programme in Dynamic Molecular Cell Biology (203959/Z/16/Z) at the University of Bristol. M.B. is supported by a British Heart Foundation on a PhD Studentship FS/20/5/34973. M.A. and M.L.J. are supported by the British Heart Foundation 4-year PhD Programme in Integrative Cardiovascular Sciences (FS/4yPhD/F/20/34125). AWP is a Wellcome Trust Investigator (219472/Z/19/Z) and is supported by grants from the British Heart Foundation (SP/F/21/150023; PG/21/10760; FS/19/53/34887). Work in the lab of I.H. is supported by the UKRI Biotechnology and Biosciences Research Council (BB/X017176/1) and the National Centre for Replacement, Refinement and Reduction of Animals in Research. D.M.P. has been supported by the British Heart Foundation through a Career Re-Entry Fellowship (FS/14/18/3071) and a current Intermediate Basic Science Research Fellowship FS/IBSRF/23/25156. This work was also supported by a Lister Institute of Preventative Medicine Fellowship to M.P.D., and work in his lab is supported by the UKRI Biotechnology and Biosciences Research Council (BB/W005581/1). We thank Andrew P. Carter and Camilla V. Santos (MRC-LMB) for their comments on and support of the project and Michael Way (Francis Crick Institute) and Edward H. Egelman (University of Virginia) for helpful discussions. We thank Edward Morris (University of Glasgow) for help and collaboration with IMAGIC software. We acknowledge access and support of the GW4 Facility for High-Resolution Electron Cryo-Microscopy, funded by the Wellcome Trust (202904/Z/16/Z and 206181/Z/17/Z) and BBSRC (BB/R000484/1). We acknowledge Diamond for access and support of the cryoEM facilities at the UK national electron Bio-Imaging Centre (eBIC), proposals BI25452-24, BI32707-3, BI32707-7, BI32707-12, and BI32707-14, funded by the Wellcome Trust, MRC, and BBSRC.

## Author contributions

Conceptualisation: C.T., P.V., D.M.P., A.W.P., I.H., M.P.D.; Formal analysis: C.T., M.B., J.M., I.H., P.V., A.W.P., D.M.P., M.P.D.; Investigation: C.T., M.B., J.M., I.H., P.V., A.W.P., D.M.P., M.P.D.; Methodology: C.T., M.B., M.F.A., M.L.J., J.M., U.B.; Writing—original draft: C.T., M.P.D.; Writing–reviewing and editing: All authors.

## Competing interests

The authors declare no competing interests.
