## [Peer Review File · Nature Communications]

Reviewers' Comments:

Reviewer #1:

Remarks to the Author:

In their manuscript „CryoET reveals actin filaments within platelet microtubules“ Tsuji et al. characterize actin filaments in the microtubule lumen based on their previous findings. They show that under chemical treatment and in vivo cofilin-bound actin is the predominant form in the MT lumen. However, they also demonstrate that in human platelets, without any special treatment, F-actin can be found in the MT lumen.

The manuscript is well written and very concise. I wonder if the authors could try to better characterize the microtubules that harbor actin and also provide some more quantitative data on the actin.

Specifically I wonder if the authors could from their cryoET data quantify the protofilament number of the microtubules that contain actin. Also, it would be great to see if actin occurs in microtubules with posttranslational modifications. The authors could check by fluorescence if there are more filaments in acetylated, tyrosinated or polyglutamylated microtubules. The hypothesis for now is that actin polymerizes/enters close to microtubule defects. Is it possible to detect those and quantify the distance of actin to the defects?

What are the length distributions of the different classes of actin filaments?

Do the authors have enough data to perform subtomogram averaging?

Some minor points to the figures:

Figure 1: In panel A I would show the untreated cells in the same magnification as the treated ones. Are there never any MT actin filaments in the untreated or does this happen even if it is very rare?

Figure 3: Do the authors have data from fluorescence microscopy for this? Are there preferred regions for the MT-actin in platelets? Do the authors have helical reconstructions and power spectra for the cofilin-actin similar to their HAP data?

Reviewer #2:

Remarks to the Author:

Functional cooperation between the different filaments of the cytoskeleton is an important aspect of cell activity but the molecular mechanisms by which this is achieved remain poorly understood. The considerable cell volume within the lumen of microtubules has long fascinated cell biologists and numerous studies have reported a range of microtubule luminal content. The more recent fascinating observation by the current authors of actin filaments inside microtubules opened up new perspectives on inter-filament regulation.

The straightforward manuscript by Tsuji et al extends the authors' previous work and studies from other groups and presents i) further cryo-ET data and analysis of microtubules in kinesin-treated HAP1 cells, thereby allowing more detailed structural analysis of these filament populations, ii) in vitro reconstitution experiments, iii) cryo-FIB thinned human platelets. The experiments are systematically performed and presented, the data are robustly analysed and will be of interest to the broad cytoskeleton community. The origin and functional significance of the presence of actin (and other filaments) within microtubules remains mysterious, but such investigations are firmly the target of future studies.

A number of points could be usefully clarified/expanded in the current manuscript:

1. While substantial circumstantial evidence supports the conclusion that the Class I filaments observed in HAP1 cells by the authors are cofilin-bound F-actin, the resolution of the reconstruction presented is not sufficiently detailed to support that conclusion directly. Is anything known about the expression patterns of cofilin and related proteins in the HAP1 cells that could add weight to this assertion?

2. Is there any variation in the protofilament number of the microtubules in any of the datasets and if yes, is there any correlation with the presence/absence/type of inter-luminal filaments?
3. Obviously the focus of Tsuji et al is the microtubules that do contain filaments, but what fraction of the total microtubules observed don't? This information is provided for the in vitro experiments but not the other datasets (that I could find) and would be important to include. Similarly, it would be beneficial for the more general reader to be explicit about the range of organisms/systems (e.g. PMID:17562819, PMID:34698018, PMID: 36869034) where actin filaments within microtubules have so far not been reported.
4. The demonstration that intra-luminal actin can be reconstituted in vitro, albeit infrequently, is an important step. The authors state the specific goal of these experiments was "To define basic requirements for the incorporation of F-actin into microtubules", but it is not completely what these basic requirements were found to be, other than the presence of polymerisation-competent actin and tubulin. How many experimental parameters were varied? "Significant mechanical stress" is proposed as a mechanism of generating intra-luminal actin in cells. Do the authors consider there is equivalent stress in the in vitro experiments?
5. Can anything mechanistic be learnt about the origin of intra-luminal filaments in platelet microtubules from the context in which they are observed - for example, bundles vs free microtubules, proximity to organelles etc?

Reviewer #3:

Remarks to the Author:

In their paper "CryoET reveals actin filaments within platelet microtubules", Tsuji et al. expand on previous observations of actin with the lumen of microtubules, demonstrating that this phenomenon not likely to be confined to cultured cells under the influence of cytoskeleton-affecting drugs. The two previous reports that detailed this (Paul et al., JCB 2020; Santos et al., EMBO Rep. 2023) left open the possibility that inter-luminal actin was not physiologically relevant due to the extensive manipulations of cells needed to obtain thin enough samples for cryo-electron tomography (cryo-ET). In this current report, the authors present two pieces of evidence that this not case. First, instead of chemical manipulations, they employed focused ion-beam (FIB) milling to thin minimally perturbed platelet cells from human donors, where they observed actin-filled microtubules in-situ. Second, they reconstitute the phenomenon in-vitro, copolymerizing actin and microtubules and finding that some microtubules contained actin using cryo-ET. In addition, also recapitulate the previous report that found that cofilin-decorated actin was the most prominent form of inter-luminal actin, in platelets, in vitro, and in the previously-used chemically manipulated HAP1 cells by helical averaging and layer line analysis.

In all, this we found that this report to be clear in its observations and concise in its details. After the previous reports, it was not clear if inter-MT lumen actin was a general phenomenon or just an experimental fluke. This work conclusively shows that it is not, which will likely incentivize other researchers to look for these same type of structures in their data and consider what influence they might have physiologically. However, we do have some minor concerns, related to quantification of the data and expansion upon the image analysis procedures.

Minor concerns:

1. Density from a helical reconstruction was presented, but the method it was obtained was not explained. Please include the detailed procedures (software used, number of segments incorporated, etc.) in the Methods section. Additionally, the reconstruction should be deposited in a publicly available repository, e.g. the EMDataBank, but no Data Availability statement is included.
2. The study would be strengthened by rigorous quantification of the observations from the in-vitro cryo-ET study to see if there is a preference for cofilactin. It has previously been speculated that cofilin decoration promoted actin presence in the microtubule lumen, which is consistent with the observation in this paper that about double the number of MTs in reactions with cofilin had actin

compared to without (5% vs 2%) as alluded to in the text. A detailed quantification should be presented in support of this (number of tomograms, number of filaments observed across different categories, etc.).

3. The study would additionally be strengthened by quantification of the platelet data presented in Figure 3. How many lamella / cells were analyzed? How many microtubules were observed, and what fraction contained cofilactin? The quantifications presented by Santos et al. in their 2023 paper provide a good example of a rigorous approach.

Additional comments:

-In the abstract / introduction, the authors state that actin microtubule crosstalk is "essential for many cellular processes" without providing any additional context. It could be useful to expand this with some specific examples in the intro. Furthermore, choice of the word "essential" is a bit problematic, as most documented examples of crosstalk are from observational studies of phenomena in cells, disruption of individual crosslinkers whose precise functions are difficult to pinpoint, or in vitro reconstitution experiments. While cytoskeletal crosstalk is certainly important and historically understudied, it is useful to be precise about where the field stands.

-In the introduction, the authors further state that the discovery of intraluminal actin "challenges the framework" of previously proposed cytoskeletal crosstalk mechanisms mediated by signaling pathways or crosslinkers. This is inaccurate, as the discovery of intraluminal actin does nothing to falsify those alternative mechanisms, for which there is very strong evidence. It would be more accurate to state that the discovery of intraluminal actin provides an additional mechanism by which cytoskeletal crosstalk can occur.

We are very grateful to the reviewers for the time and effort put into providing positive and constructive feedback for this work. Our point-by-point response is provided below.

REVIEWER COMMENTS

Reviewer #1 (Remarks to the Author):

In their manuscript „CryoET reveals actin filaments within platelet microtubules” Tsuji et al. characterize actin filaments in the microtubule lumen based on their previous findings. They show that under chemical treatment and in vivo cofilin-bound actin is the predominant form in the MT lumen. However, they also demonstrate that in human platelets, without any special treatment, F-actin can be found in the MT lumen.

The manuscript is well written and very concise. I wonder if the authors could try to better characterize the microtubules that harbor actin and also provide some more quantitative data on the actin.

Specifically I wonder if the authors could from their cryoET data quantify the protofilament number of the microtubules that contain actin.

We thank the reviewer for the suggestion and we have quantified the number of protofilaments. We analysed ten microtubules containing filaments in our platelet dataset and the same number of microtubules not containing filaments, by performing rotational averaging (Atherton et al. JCS 2022; [10.1242/jcs.259234](https://doi.org/10.1242/jcs.259234)). We focussed on the platelet set because this is clearly more physiologically relevant than the HAP1 cell system where we see evidence of substantial microtubule disruption (Fig. S3). We found that the microtubules in platelets consisted of 13 or 14 protofilaments (80% vs 20%) which is consistent with previous literature on platelet protofilaments (Xu and Afzelius (1988); [10.1016/0889-1605\(88\)90068-7](https://doi.org/10.1016/0889-1605(88)90068-7) and Chaaban and Brouhard (2017); [10.1091/mbc.E16-05-0271](https://doi.org/10.1091/mbc.E16-05-0271)). Whilst the relatively small FIB-milled dataset limits capacity for statistical analysis, our quantification did not suggest any differences protofilament number of microtubules which contain or do not contain filaments; we observed luminal filaments in both 13 and 14 protofilament microtubules. This is also consistent with results from Santos et al., 2023 (EMBO Reports) who also did not note any correlation between luminal filaments and protofilament number. We have added our findings to the text in lines and as a new supplementary figure (Fig. S4).

The new text reads “Platelet microtubules have been reported to contain 13-15 protofilaments^{24,26}. To ask whether the presence of a luminal filament is associated with a difference in protofilament number we performed rotational averaging of microtubule from our platelet tomographic reconstructions¹². In both cases (n=10), most microtubules (70-80%) were composed of 13 protofilaments, although we also observed 14 protofilament microtubules. Thus in platelets, and consistent with²⁷, presence of luminal filaments does not appear to correlate with protofilament number (Fig. S4).”

Also, it would be great to see if actin occurs in microtubules with posttranslational modifications. The authors could check by fluorescence if there are more filaments in acetylated, tyrosinated or polyglutamylated microtubules.

We agree with the reviewer that this is an interesting and important question. We have thought a lot about how to address it, but we do not think that this is tractable at this point using the approaches that we have. The challenge lies in identifying which microtubules are post-translationally modified in cryoET experiments (which is the only system where the presence of luminal actin can currently be unambiguously observed and distinguished from actin that is outside the microtubule). This precludes fixation and immunostaining, which is the best way to observe PTMs, but also is very much limited by the resolution limits of conventional light microscopy, which means that microtubules within a bundle

of 200-400 nm diameter are difficult to resolve. Going forwards, there may be some potential for super-resolution imaging coupled with cryo-CLEM, or possibly expansion microscopy, but this will be a long-term technical commitment.

To reflect the reviewer's comment, we have added a line in the discussion to highlight that this should be a focus for future work.

This new text reads “...It will be important to consider factors that may control its incorporation or function such as microtubule posttranslational modification.”

The hypothesis for now is that actin polymerizes/enters close to microtubule defects. Is it possible to detect those and quantify the distance of actin to the defects?

We have looked at this particularly closely for the platelet dataset but it is not obvious that luminal actin filaments are associated with any defects that we can see. We want to explore when the luminal actin is incorporated (for example, as discussed during platelet biogenesis from megakaryocytes when microtubules are 'looped out') to understand whether defects exist there that are subsequently repaired, but at this point, we cannot make a strong statement on this.

What are the length distributions of the different classes of actin filaments?

We have measured the length distribution for the HAP1 and platelet datasets. There are now presented in Figure 1 and Figure 3.

Do the authors have enough data to perform subtomogram averaging?

We have attempted subtomogram averaging on all three classes of luminal filaments. However, we did not have enough data to obtain clear insights. This is an important goal for our ongoing work – particularly for the Class II filament which appears highly ordered and so is more promising, but this will require significant investment in time and effort for data acquisition and processing.

Some minor points to the figures:

Figure 1: In panel A I would show the untreated cells in the same magnification as the treated ones. Are there never any MT actin filaments in the untreated or does this happen even if it is very rare?

Figure 1 has been modified to show the untreated HAP1 cell with the same magnification as kinesore treated HAP1 cells.

We were able to image the kinesore treated HAP1 cells using cryo-ET due to the thin nature of the cell projections, which enabled us to find microtubule luminal filaments. Untreated HAP1 cells are too thick to image using standard cryo-ET. Whilst we are interested in exploring the presence of microtubule luminal filaments in untreated HAP1 cells, our priority has to move this to a primary cell (platelet) system (without treatments) rather than this extensively genetically modified line. In the platelets, they are indeed less frequent than in kinesore treated HAP1 cells.

Figure 3: Do the authors have data from fluorescence microscopy for this? Are there preferred regions for the MT-actin in platelets?

As discussed above, this is challenging because the only way to unambiguously observe filaments inside the microtubule is using cryoET. At this point, we can say that they are found in microtubules with a bundled morphology consistent with the marginal band as well as some apparently more isolated microtubules.

Do the authors have helical reconstructions and power spectra for the cofilin-actin similar to their HAP data?

We thank the reviewer for this suggestion. We have included a helical reconstruction of a platelet filament in Figure 3 and a power spectrum in the Supplementary Figure 4.

Reviewer #2 (Remarks to the Author):

Functional cooperation between the different filaments of the cytoskeleton is an important aspect of cell activity but the molecular mechanisms by which this is achieved remain poorly understood. The considerable cell volume within the lumen of microtubules has long fascinated cell biologists and numerous studies have reported a range of microtubule lumenal content. The more recent fascinating observation by the current authors of actin filaments inside microtubules opened up new perspectives on inter-filament regulation.

The straightforward manuscript by Tsuji et al extends the authors' previous work and studies from other groups and presents i) further cryo-ET data and analysis of microtubules in kinesin-treated HAP1 cells, thereby allowing more detailed structural analysis of these filament populations, ii) *in vitro* reconstitution experiments, iii) cryo-FIB thinned human platelets. The experiments are systematically performed and presented, the data are robustly analysed and will be of interest to the broad cytoskeleton community. The origin and functional significance of the presence of actin (and other filaments) within microtubules remains mysterious, but such investigations are firmly the target of future studies.

A number of points could be usefully clarified/expanded in the current manuscript:

1. While substantial circumstantial evidence supports the conclusion that the Class I filaments observed in HAP1 cells by the authors are cofilin-bound F-actin, the resolution of the reconstruction presented is not sufficiently detailed to support that conclusion directly. Is anything known about the expression patterns of cofilin and related proteins in the HAP1 cells that could add weight to this assertion?

We agree with the reviewer's point here. Whilst the classic cofilin-bound filament morphology, 3D reconstruction, and similarly to that observed by Santos *et al.* and other cryoET studies, do very strongly support the proposition that the Class I filament is cofilin-bound F-actin, we have rephrased our conclusion at the end of the first results section to state that '*these data strongly support the proposition that the Class I filament is cofilin-bound F-actin*'. In light also of comment 4. from the reviewer on the rationale for the *in vitro* reconstitution experiments, we have slightly changed the emphasis of this section, as a further experimental test of this, by seeking to recapitulate our *in situ* observations using defined purified components. We hope that the reviewer agrees that this improves the logic through the manuscript and helps to build a robust case.

The reviewer makes a good point on the identity of the cofilin family member. Analysis of published transcriptomic data from HAP1 cell line used here ([10.1101/gr.177220.114](https://doi.org/10.1101/gr.177220.114)) indicates that CFN1 is the predominant form at around 330 TPM. The typically muscle specific CFN2 is expressed at very low levels at 45 TPM. Destrin/ADF is around 100 TPM. As such, *in situ*, whilst there could be a mixture of cofilin proteins, CFN1 is the most likely candidate. *In vitro*, we focussed on CFN1. We added the point that CFN1 is the most highly expressed family member to manuscript and cited the above paper.

2. Is there any variation in the protofilament number of the microtubules in any of the datasets and if yes, is there any correlation with the presence/absence/type of inter-lumenal filaments?

We thank the reviewer for the suggestion and we have quantified the number of protofilaments. We analysed ten microtubules containing filaments in our platelet dataset and the same number of microtubules not containing filaments, by performing rotational averaging (Atherton et al. JCS 2022; 10.1242/jcs.259234). We focussed on the platelet set because this is clearly more physiologically relevant than the HAP1 cell system where we see evidence of substantial microtubule disruption (Fig. S3). We found that the microtubules in platelets consisted of 13 or 14 protofilaments (80% vs 20%) which is consistent with previous literature on platelet protofilaments (Xu and Afzelius (1988); [10.1016/0889-1605\(88\)90068-7](https://doi.org/10.1016/0889-1605(88)90068-7) and Chaaban and Brouhard (2017); [10.1091/mbc.E16-05-0271](https://doi.org/10.1091/mbc.E16-05-0271)). Whilst the relatively small FIB-milled dataset limits capacity for statistical analysis, our quantification did not suggest any differences protofilament number of microtubules which contain or do not contain filaments; we observed luminal filaments in both 13 and 14 protofilament microtubules. This is also consistent with results from Santos et al., 2023 (EMBO Reports) who also did not note any correlation between luminal filaments and protofilament number. We have added our findings to the text in lines and as a new supplementary figure (Fig. S4).

The new text reads “Platelet microtubules have been reported to contain 13-15 protofilaments ^{24,26}. To ask whether the presence of a luminal filament is associated with a difference in protofilament number we performed rotational averaging of microtubule from our platelet tomographic reconstructions ¹². In both cases (n=10), most microtubules (70-80%) were composed of 13 protofilaments, although we also observed 14 protofilament microtubules. Thus in platelets, and consistent with ²⁷, presence of luminal filaments does not appear to correlate with protofilament number (Fig. S4).”

3. Obviously the focus of Tsuji et al is the microtubules that do contain filaments, but what fraction of the total microtubules observed don't? This information is provided for the in vitro experiments but not the other datasets (that I could find) and would be important to include. Similarly, it would be beneficial for the more general reader to be explicit about the range of organisms/systems (e.g. PMID:17562819, PMID:34698018, PMID: 36869034) where actin filaments within microtubules have so far not been reported.

We appreciate the reviewer's point. This has been included in Figures 1 and 3 and Supplementary Figure 2.

For the HAP1 dataset, 26 tomograms were quantified which included a total of 144 microtubules of varying lengths. The sum total of microtubule lengths that were measured was 82.7 μm . 72.6% of the measured microtubule lengths were filament free (although most of these contained globular densities), whereas 27.4% contained luminal filaments. The total length of the measured filaments was 22.7 μm - out of this, 70.6% were class I, 13.1% class II, 7.1% class III and 0.5% were F-actin and 8.6% were unclear in their morphology.

We agree that it is helpful to broaden the discussion to consider papers (including those mentioned by the reviewer), that have not observed luminal filaments. We have modified the discussion paragraph to reflect this.

4. The demonstration that intra-luminal actin can be reconstituted in vitro, albeit infrequently, is an important step. The authors state the specific goal of these experiments was “To define basic requirements for the incorporation of F-actin into microtubules”, but it is not completely what these basic requirements were found to be, other than the presence of polymerisation-competent actin and tubulin. How many experimental parameters were varied? “Significant mechanical stress” is

proposed as a mechanism of generating intra-luminal actin in cells. Do the authors consider there is equivalent stress in the *in vitro* experiments?

We agree that 'define basic requirements' was not the best choice of phrase here. We were aiming to understand whether the formation of these novel structures, at a fundamental level, requires many factors in the complex cellular environment, or whether actin and tubulin (and cofilin) are sufficient. We were also aiming to add weight to the notion that the short-crossover filaments observed in HAP1 cells and platelets are indeed formed of cofilin-bound F-actin by recapitulating our *in situ* observations *in vitro* using defined components. As outlined for point 1, we have changed the introduction to this section to better articulate this.

It is interesting to consider whether the *in vitro* polymerisation conditions used here do introduce any mechanical stress. This seems unlikely, although there are obvious breaks and gaps in the microtubule lattice in some cases (Figure 2, panel E). Perhaps could be the reason for the relatively infrequent incorporation in these conditions. Going forwards, it would be interesting to see whether we can incorporate motor driven sliding/lattice damage into this system and analyse its effects.

5. Can anything mechanistic be learnt about the origin of intra-luminal filaments in platelet microtubules from the context in which they are observed - for example, bundles vs free microtubules, proximity to organelles etc?

We would very much like to explore this. It isn't obvious from the data we have – for example within microtubules of the marginal band morphology, apparently fully intact adjacent microtubules (Figure 3B/Supplementary Movie 2) may or may not have luminal filaments. This will be a priority, but it will require us to expand our datasets considerably.

Reviewer #3 (Remarks to the Author):

In their paper "CryoET reveals actin filaments within platelet microtubules", Tsuji et al. expand on previous observations of actin with the lumen of microtubules, demonstrating that this phenomenon not likely to be confined to cultured cells under the influence of cytoskeleton-affecting drugs. The two previous reports that detailed this (Paul et al., JCB 2020; Santos et al., EMBO Rep. 2023) left open the possibility that inter-luminal actin was not physiologically relevant due to the extensive manipulations of cells needed to obtain thin enough samples for cryo-electron tomography (cryo-ET). In this current report, the authors present two pieces of evidence that this not case. First, instead of chemical manipulations, they employed focused ion-beam (FIB) milling to thin minimally perturbed platelet cells from human donors, where they observed actin-filled microtubules *in-situ*. Second, they reconstitute the phenomenon *in-vitro*, copolymerizing actin and microtubules and finding that some microtubules contained actin using cryo-ET. In addition, also recapitulate the previous report that found that cofilin-decorated actin was the most prominent form of inter-luminal actin, in platelets, *in vitro*, and in the previously-used chemically manipulated HAP1 cells by helical averaging and layer line analysis.

In all, this we found that this report to be clear in its observations and concise in its details. After the previous reports, it was not clear if inter-MT lumen actin was a general phenomenon or just an experimental fluke. This work conclusively shows that it is not, which will likely incentivize other researchers to look for these same type of structures in their data and consider what influence they might have physiologically. However, we do have some minor concerns, related to quantification of the data and expansion upon the image analysis procedures.

Minor concerns:

1. Density from a helical reconstruction was presented, but the method it was obtained was not explained. Please include the detailed procedures (software used, number of segments incorporated, etc.) in the Methods section. Additionally, the reconstruction should be deposited in a publicly available repository, e.g. the EMDataBank, but no Data Availability statement is included.

Our apologies for missing this information. Methods have been updated. Helical reconstruction maps are deposited in the Electron Microscopy Data Bank with accession code EMD-50350 for the HAP1 helical reconstruction and EMD-50351 for a new platelet helical reconstruction. This has been added in the figure legends and new Data Availability statement.

2. The study would be strengthened by rigorous quantification of the observations from the in-vitro cryo-ET study to see if there is a preference for cofilactin. It has previously been speculated that cofilin decoration promoted actin presence in the microtubule lumen, which is consistent with the observation in this paper that about double the number of MTs in reactions with cofilin had actin compared to without (5% vs 2%) as alluded to in the text. A detailed quantification should be presented in support of this (number of tomograms, number of filaments observed across different categories, etc.).

We appreciate the reviewers point. To address this, we have acquired an additional dataset (now three independent polymerisation and freezing sessions for both conditions). The results are reported in Supplementary Figure 2. Substantially, the conclusion remains the same, we see a slightly higher (but not statistically significant difference) frequency of incorporation in cofilin-actin samples – if there is an effect of cofilin in this system, it is small, but this does not exclude an important role for cofilin in cells. We have included the additional information requested in the methods and figure legends.

3. The study would additionally be strengthened by quantification of the platelet data presented in Figure 3. How many lamella / cells were analyzed? How many microtubules were observed, and what fraction contained cofilactin? The quantifications presented by Santos et al. in their 2023 paper provide a good example of a rigorous approach.

Thank you. As requested, we have provided additional quantification of the platelet data. Across 2 datasets, 28 tomograms were quantified corresponding to 23 lamella. These contained 63 microtubules which were visible. The measured total length of the microtubules were 63.7 μm of which 95.5% were filament free (although many containing globular particles and density that was difficult to define) and 4.5% contained clear luminal filaments. The total length of filaments inside the microtubule lumen was 2.85 μm , of which 73.1% was class I cofilin-bound actin and 26.9% had an F-actin like morphology. We have added this information to the manuscript in Figure 3.

Additional

comments:

-In the abstract / introduction, the authors state that actin microtubule crosstalk is “essential for many cellular processes” without providing any additional context. It could be useful to expand this with some specific examples in the intro. Furthermore, choice of the word “essential” is a bit problematic, as most documented examples of crosstalk are from observational studies of phenomena in cells, disruption of individual crosslinkers whose precise functions are difficult to pinpoint, or in vitro reconstitution experiments. While cytoskeletal crosstalk is certainly important and historically understudied, it is useful to be precise about where the field stands.

We thank the reviewer for pointing this out. We have modified the abstract and introductory paragraph to reflect these comments.

In the introduction, the authors further state that the discovery of intraluminal actin “challenges the framework” of previously proposed cytoskeletal crosstalk mechanisms mediated by signaling pathways or crosslinkers. This is inaccurate, as the discovery of intraluminal actin does nothing to falsify those alternative mechanisms, for which there is very strong evidence. It would be more accurate to state that the discovery of intraluminal actin provides an additional mechanism by which cytoskeletal crosstalk can occur.

We appreciate the reviewer’s point. We have modified the introduction as the reviewer suggested.

Reviewers' Comments:

Reviewer #1:

Remarks to the Author:

In their revised manuscript the authors have provided additional data to support their findings and have sufficiently discussed why some experiments, especially fluorescence imaging of luminal actin, are very difficult to conduct.

I therefore recommend publications of the revised manuscript.

Reviewer #2:

Remarks to the Author:

I am satisfied with the corrections to this manuscript. Congratulations to the authors on a very nice and well presented study.

Reviewer #3:

Remarks to the Author:

In their revised manuscript, Tsuji et al. were highly responsive to the comments raised by ourselves and the other reviewers. We believe the additional quantification, as well as addressing data availability, expanding the methods, and adjusting various claims in the text, has substantially strengthened the paper. We believe it is now suitable for Nature Communications, and should be accepted without further delay.